# A Unified Pathogenesis of Allergic Diseases; The Protein–Homeostasis–System Hypothesis

**DOI:** 10.3390/ijms26178358

**Published:** 2025-08-28

**Authors:** Younhee Ko, Kyung-Yil Lee

**Affiliations:** 1Division of Biomedical Engineering, Hankuk University of Foreign Studies, Yongin City 17035, Republic of Korea; younko@hufs.ac.kr; 2Junglock Biomedical Institute, Deajeon 34886, Republic of Korea; 3College of Medicine, The Catholic University of Korea, Seoul 06591, Republic of Korea

**Keywords:** allergy, immunology, eosinophil, mast cell, immunoglobulin E, etiology, pathophysiology, protein–homeostasis–system hypothesis

## Abstract

The etiology and pathophysiology of allergic diseases remain incompletely understood. Current immunological paradigms, while insightful, often fall short in fully elucidating the mechanisms underlying allergic and autoimmune disorders. Under the protein–homeostasis–system (PHS) hypothesis, allergic diseases have etiological substances, and immune reactions against them are responsible for clinical manifestations of allergic diseases. The etiological substances are mainly external in origin and very small with each biochemical property and react to target cells in various organ tissues. Eosinophils, mast cells, and immunoglobulin Es as major immune effectors in allergic diseases control toxic substances according to the chemical or biochemical properties of these substances. Mast cells in the central nervous system may be associated with allergic episodes through connection to peripheral mast cells, and this connection is proposed as the mast cell-associated network. The toxic and/or bioactive proteins/peptides and other non-protein substances, which are derived from injured cells caused by allergic reactions, induce activation of adaptive and innate immune components for controlling the substances. New insights into the etiology and pathophysiology of allergic diseases are introduced with the PHS hypothesis.

## 1. Introduction

Diseases are traditionally classified by organs and further subdivided into categories such as allergic and autoimmune diseases. Diseases in both categories are caused by abnormal immune reactions and affect multiple organs with varied clinical manifestations. Asthma and atopic dermatitis (AD) among allergic diseases, and rheumatoid arthritis (RA) and systemic lupus erythematosus (SLE) among autoimmune conditions are representative examples. The increased incidence or prevalence rates of these diseases in industrialized societies depend on environmental and genetic factors [1,2]. In addition, the epidemiology and severity of allergic diseases are changing over time in each country [3].

Most researchers in biomedical fields agree that every disease has an etiology, but current methods and the complexity of diverse pathophysiological processes of the disease do not always identify the causes. In science, etiology refers to the specific substances or factors responsible for disease onset. There are various substances that can be harmful to the host, including elements, monoamines, chemicals, peptides, proteins, and other biochemicals. The effect of these small molecules occurs through their interaction with receptors expressed on or within affected cells known as target cells. Current immunological models have limitations to explain the pathophysiology of allergic and autoimmune diseases as well as other diseases including infectious diseases, organ-specific diseases, and cancer. All biological organisms, from unicellular to complex multicellular organisms, could be considered as an integrated biosystem, thus it is reasonable to propose that there exists a unifying mechanism(s) that controls biological activities including embryonal development, maintaining homeostasis of a life, and biochemical and immune reactions against insults. We refer to this conceptual system as the protein–homeostasis–system (PHS). Although similar concepts in the organismic systems biology based on ‘organicism’ have been introduced [4], there have been no theories or hypotheses for a unified pathophysiology of the diseases.

The PHS hypothesis proposes that all diseases involve etiological substances (or causal events such as protein deficiencies) whose size and biochemical characteristics determine the corresponding immune responses and subsequent disease-onset and phenotype of diseases. Accordingly, clinical symptoms and signs of diseases appear during the process of the immune response to these substances. Pathological findings, particularly in the early stages of disease, offer critical clues to these underlying etiological substances. Failure of the immune system to effective control of such substances leads to persistent inflammation and the development of chronic inflammatory or autoimmune diseases. We have previously applied the PHS hypothesis to a range of clinical entities, including infectious diseases such as influenza and COVID-19, immune-mediated syndromes such as Kawasaki disease (KD) and multi-systemic inflammatory syndrome in children (MIS-C), organ-specific diseases such as all kidney diseases and central nervous system (CNS) diseases containing genetic diseases and cancers [5,6,7,8,9,10,11,12,13].

In this manuscript, we extend the PHS framework to allergic diseases, proposing a unified pathophysiology that integrates the roles of etiological substances and their immune effectors in determining disease phenotype, progression, and outcomes.

## 2. A Brief History of Immunology and Allergy

Human hosts coexist with diverse strains in their microbiota in a symbiotic relationship; however, under certain conditions, commensal organisms (here referred to as internal pathogens), as well as exogenous (or external) pathogens originating from other species or environments, can invade the host and cells of the host, leading to infectious and infection-related immune-mediated diseases. Although allergic diseases such as asthma have existed for thousands of years, their prevalence or incidence have significantly increased, particularly in response to industrialization and westernization over the past few centuries [14]. People in modern society are exposed to a vastly different environment, characterized by westernized diets rich in red meats and animal fats, and widespread exposure to synthetic chemicals including pharmaceuticals, petrochemicals, food additives, and genetically modified products.

Explosive outbreaks of infectious disease, including those affecting livestock and agricultural plant, and zoonotic diseases, have appeared along with growth of human society, and infectious diseases became the center of medical research since the late 19th century. Concurrently, as socioeconomic structures evolved, new health challenges emerged; metabolic diseases linked to overnutrition such as obesity and diabetes now constitute major public health burdens [15]. With greater health awareness among populations in developed societies, modern medicine has advanced considerably to meet these needs; however, unresolved challenges in understanding and managing complex diseases remain [16].

Experienced clinicians throughout medical history have consistently observed two fundamental principles: first, that individuals with the same infectious disease experience markedly different clinical outcomes; and second, that recovery from an infectious disease often confers protection against reinfection. These two observations could be defined as ‘natural healing power’ and ‘immunity’, respectively. Natural healing power (original meaning of vis medicatrix naturae [17]) can refer to the self-defense system that the organism’s innate capacity for defense and cellular restoration following exposure to harmful agents such as toxins, microbes, and physical trauma. Although researchers do not use the term natural healing power much due to its holistic and philosophical connotations, the concept of immunity based on the infection could be included in the natural healing power. Conversely, the broad sense of immunity can replace natural healing power, since the immune systems of the host work for every disease, including infectious diseases, allergic diseases, cancer, and tissue repair.

The immunology begun in the late 19th century with the discovery of pathogens causing infectious diseases and the development of vaccines against pathogens. Early models pointed out the roles of antibodies and phagocytic cells in defending the host against infectious agents such as pathogens and toxins. The acceptance of ‘germ theory’ emphasized the idea that diseases have specific causal pathogens or causal substances [18]. Over time, immunology has evolved in parallel with advances in molecular biology and biotechnology [19], and most researchers in immunological and allergic fields accept the concept that the immune system has an inherently defensive mechanism to protect the host through an innate capacity to distinguish between benign and toxic, or self and non-self.

The conceptual linkage between immunology and allergology was established when it became evident that immune responses could also contribute to noninfectious diseases, and that, in some cases, these responses could be detrimental to the host, as seen in hypersensitivity reaction and anaphylaxis [20]. In 1906, Austrian pediatrician Clemens von Pirquet introduced the term ‘allergy’ to describe a biological phenomenon in which immune responsiveness manifests either as protective immunity or as pathological hypersensitivity, via a common underlying mechanism [21]. Importantly, he proposed that there were causative substances, named ‘allergens’ in the immune responsiveness including not only infectious agents and toxins but also other substances such as immune complexes. Pirquet’s theory was based on clinical observations of serum sickness, smallpox vaccination, and tuberculosis and framed within an organismic view [22]. The substances were difficult to validate experimentally at the time and even in present time. Being different to his initial concept, the term ‘allergy’ has since evolved to denote harmful or exaggerated immune responses, primarily of the hypersensitivity type [23]. The concept of ‘atopy’ emerged in the 1920s to describe inherited or constitutional predispositions toward hypersensitive reactions, particularly in asthma and hay fever. The causative agents were termed ‘atopic reagins’, and some of which were found to be transferable via serum in experimental allergic diseases [24]. Following the discovery of immunoglobulin E (IgE), it became widely accepted that IgE mediates immediate-type (type I) hypersensitivity, as suggested by Gell and Coombs, and serves as a hallmark of atopic diseases [25,26]. However, phenotypically similar conditions such as non-IgE-mediated asthma, rhinitis, or eczema have also been observed, suggesting that distinct etiological substances activate common immune pathways. During the development of allergology, the definition and pathogenesis of allergic diseases, including asthma and food allergy, could not be clearly defined and have been revised as well as other medical disciplines [27,28].

## 3. Evolutionary Aspects of the Immune System: Immunological Memory and Cross-Reactivity

Multicellular organisms, including humans, are believed to have evolved from unicellular ancestors. Even single-cell organisms such as bacteria, yeasts, and fungi possess primitive defense systems against viruses (e.g., bacteriophages) and toxic substances that are very small. These include immune-related proteins such as restriction enzymes (e.g., CRISPR-Cas systems), antimicrobial peptides, and biochemical agents like penicillin [29]. Despite their far earlier evolutionary origin compared to mammals, plants and insects retain only innate immune systems [30,31]. In insects such as the fruit fly, the major immune system consists of phagocytes with Toll receptors and immune protein systems against external insults, which have immunological memory and specificity [31]. In mammals, innate immune cells such as phagocytes can effectively eliminate extracellular pathogens, including bacteria and viruses, circulating in the bloodstream. However, these cells are limited in their capacity to eliminate pathogens within cells, such as viruses and intracellular pathogens, or toxic substances derived from host cells. From an evolutionary perspective, if all toxic threats were exclusively external, the development of an adaptive immune system may not have been necessary. The emergence of complex organ systems in mammals, which are composed of each specialized cell expressing distinct receptors and producing distinct peptides and proteins, may necessitate a more sophisticated immune architecture. Pathogen-associated molecular patterns (PAMPs) and damage-associated molecular patterns (DAMPs), originating from pathogens and self-cells, respectively, can induce similar immune reactions through pattern recognition receptors (PRRs) such as Toll-like receptors (TLRs) and cytosolic receptors including NOD-like receptors [32]. These receptors function analogously to Toll receptors in insects. Thus, the adaptive immune system in mammals may have evolved primarily to manage internal threats, such as toxic substances from the self-cells and byproducts from dysregulated microbiota, rather than solely combating external pathogens. It is an acceptable notion that a variety of biological mechanisms such as apoptosis, autophagy, extracellular traps of granulocytes, and epigenetic alterations (e.g., DNA methylation) observed in pathological conditions reflect the host control system’s effort to reduce such internal toxins at least in part.

### 3.1. Immunological Memory

Immunological memory, which refers to the capacity of the immune system to mount an enhanced response upon re-exposure to the same antigen, is a defining feature of adaptive immunity. This principle underlies the efficacy of vaccines, wherein repeated exposure to vaccine antigens results in more rapid and robust antibody responses. However, emerging evidence indicates that innate immune components in mammals, including neutrophils and macrophages, also possess memory-like properties. This phenomenon, named “trained immunity,” refers to the functional reprogramming of innate immune cells, leading to enhanced responses upon secondary exposure [33]. Studies have shown that progeny of infected animals exhibit heightened resistance through enhanced protection functions of neutrophils and monocytes, suggesting that a form of immune memory is heritable, and the concept is widely used to explain enigmas in immunopathogenesis of the diseases [34,35]. Unlike adaptive memory, which depends on somatic gene recombination, trained immunity appears to rely on epigenetic reprogramming of transcriptional pathways [33]. Such memory phenomena of innate immune cells may exist across immune networks including those involved in allergic inflammation. Thus, it is possible that eosinophils and mast cells have the memory responses, and allergic individuals previously sensitized to specific allergens exhibit exaggerated responses upon re-exposure, even to minimal doses, likely due to underlying memory mechanisms.

### 3.2. Cross-Reactivity

Cross-reactive antibodies (B cells) and T cell clones targeting structurally similar but distinct antigens are well-recognized features of the adaptive immune system. The theory of the “molecular mimicry” has been proposed as one of pathogenesis of autoimmune diseases [36]. In such cases, B cells and T cells that are initially activated in response to pathogen-derived antigens inadvertently recognize and respond to self-antigens due to shared structural motifs, leading to autoimmunity through the cross-reactivity. There is accumulating evidence of co-reactive antibodies, including IgE isotypes, that recognize both pollens and food allergens, as well as shared epitopes between environmental allergens and metazoan parasites [37,38]. Notably, cross-reactive Th17 responses to *Candida albicans* in the gut were implicated in the pathogenesis of acute allergic bronchopulmonary aspergillosis [39]. The findings suggest that T cells, including Th17 cells, and B cells can be activated by cross-reactive substances that exist across various environmental sources including allergens, foods, pathogens, and parasites. In adaptive immune responses, it is an acceptable notion that non-specific but functionally relevant antibodies and T cell clones are initially recruited via cross-reactivity when novel pathogenic antigens are encountered, since the production of pathogen-specific antibodies and T cell clones against protein/peptide antigens takes time for at least 3–4 days or more days after beginning of symptoms in infectious diseases [6]. This temporal delay of specific immune responses may be related to the antigen processing, repertoires of genes for T cell receptor (TCR) and B cell receptor (BCR) gene recombination, and class switching in adaptive immune responses. Cytokine storms occur in excessive activation of non-specific adaptive immune cells, especially T cells, against excess peptide antigens as shown in various conditions such as influenza, COVID-19, septic conditions, and autoimmune diseases [8,9,10,11]. Also, these phenomena can occur in the innate immune system, mirroring aspects of the trained immunity with immunological memory. For instance, eosinophils, basophils (functionally analogous to mobile mast cells), and mast cells can be activated by non-specific substances through cross-reactivity. It can be proposed that the functionally relevant clones work first, and activated cells release cytotoxic granules and inflammatory mediators upon initial exposure to etiological substances of allergic reactions. Following this initial response, most people obtain more allergen-specific effector clones controlling toxic substances through the trained immunity over time. Individuals with allergic diseases may have abnormal immune components including eosinophils or mast cells; the lack of specific or effective clones against allergens is responsible for exaggerated production of proinflammatory mediators, leading to harmful allergic reactions, like cytokine imbalance occurred by hyperactivity of nonspecific adaptive immune cells. Thus, it is expected that eosinophils and mast cells of allergic patients would work more through cross-reactivity.

The dual nature of the immune system, regarded as ‘double edge sword’ of the immune system, could be explained by the immune responses of specific vs. nonspecific clones of each immune component. The immune system of mammals has some pitfalls on controlling toxic substances specifically at once, suggesting that protective and potentially pathogenic immune responses may reflect evolutionary trade-offs that favor rapid defense at the cost of precision in select contexts.

## 4. Limitations of Modern Immunology

Modern immunology began with concepts based on infectious diseases, but it has evolved to explain the pathophysiology of nearly all diseases. Despite this broad applicability, the precise etiology and mechanistic pathways remain poorly understood in many diseases. While causal pathogens and allergens have been identified in infectious and allergic diseases, respectively, the detailed mechanisms by which these agents induce cell injury or tissue remodeling remain elusive.

Throughout the history of science, paradigm shifts have emerged when existing theories failed to fully account for the observed phenomena [40]. Immunology is no exception. Foundational models, including the self/non-self-discrimination paradigm, have undergone significant revision as a result of discoveries such as similar immune responses induced by both pathogen-derived substances (e.g., PAMPs) and endogenous molecules (e.g., DAMPs) [41]. Moreover, most infectious agents currently affecting humans, including viruses and intracellular bacteria, are often species-specific strains of the human microbiota rather than foreign invaders. In such infections, cell injury is not caused directly by pathogens but by smaller molecular substances produced during infection [8,9,10,11]. They include not only PAMPs, byproducts of pathogens such as toxins, viral capsids, and pathogen DNAs and RNAs, but also DAMPs, pathogenic proteins/peptides and biochemicals originating from self-cells, and some of which serve as inflammation trigger. Recent advances have revealed extensive heterogeneity among immune cells. T and B lymphocytes are subdivided into several functionally distinct subpopulations [42,43], but the responding adaptive immune cell subsets may be different according to each event. The studied functions of T cells in vivo and in vitro are very limited compared to those of antibodies (B cells) [6]. Newborns possess natural antibodies and T cells against xenogeneic proteins despite no previous exposure [44]. The innate immune components such as tissue macrophages and mast cells exhibit tissue-specific specialization [45,46], and it has been revealed that innate immune cells such monocytes/macrophages and natural killer (NK) cells, also have an immunological memory and specificity as previously discussed [33,34,35]. The activity of cytotoxic T cells and NK cells against virus-infected cells or cancer cells is frequently demonstrated in vitro and studied well on their mechanisms, but their efficacy in clinical settings such as cancer immunotherapy remains limited [13,47,48]. In cases where cytotoxic cells fail to eliminate infected or malignant cells properly, intracellular substances are released, potentially triggering inflammation. In progressive solid cancers, it is known that immune cells in the tumor microenvironments (TMEs), fail to prevent the proliferation of cancer cells and ultimately promote tumor progression [49]. For this issue, the PHS hypothesis proposes that cancer cells as a new biosystem communicate with the host’s immune cells and use the host’s immune components when they are infected or at risk of physicochemical insults [13]. Attempts to explain immune-mediated diseases by T cell subset imbalances, such as the Th1/Th2 paradigm or the balance of Th1, Th2, Th17 and Treg, have also faced limitations. These frameworks are not consistently observed in patients with the same disease and can be changed across disease stage [50]. These concepts, based on the ‘organicism’ like the ‘four humors theory’ which has been advocated for a long time in old medicine, are often insufficient to capture the complexity of immune pathophysiology.

It is reasonable to propose a thesis that humans possess an intrinsic control system that responds to disease, and the thesis is objectively correct. However, the biosystem, as a living entity, remains challenging to define, and the proposed control system encompasses a multitude of complex, interacting subsystems that function in vivo across various disease states. Within this thesis, each immune component, including immune cells, operates its own functions predetermined by cell fate under the regulation of the integrated control system. Consequently, an increased burden of etiological substances leads to heightened activation of the corresponding immune elements, resulting in more severe clinical manifestations. When an immune component is lacking or absent, the inflammatory response is temporarily reduced, but ultimately serious outcomes occur despite compensation by other immune components. For instance, severe T cell deficiencies often result in fatal outcomes, while B cell deficiencies, including Bruton’s agammaglobulinemia and selective IgA or IgE deficiencies, rarely leads to death but do increase susceptibility to infection, autoimmunity, and cancer. These observations support the idea that each immune component contributes differentially to disease onset and resolution. It was proposed that that protection against intracellular toxic peptides—regulated by T cells—appears more critical than control of toxic proteins handled by B cell-derived antibodies [6]. Also, not only components in adaptive immune system, but also those in innate immune system are involved in any events associated with self-cell injury and repair.

## 5. The PHS Hypothesis

The PHS hypothesis has explained the etiology and pathophysiology of the disease based on the “deductive method” with minimal contradictions. At its core, the hypothesis suggests that all diseases involve etiological substances that provoke immune responses and disease onset. The immune system, as a part of the PHS, controls or neutralizes these substances based on their molecular size and biochemical characteristics. The PHS also regulates intracellular or systemic protein deficiencies at least in part. Here, more detailed key principles of the PHS hypothesis that has been announced so far are as follows.

### 5.1. Functional Consistency of Immune Components and the Foci of Etiological Substances

The function of each component of the immune system is predetermined and operates consistently across various disease lesions. The primary roles of adaptive immune cells, T cells and B cells, are believed to be the regulation of pathogenic proteins and peptides, respectively. Peptides are commonly regarded as fragments of proteins, typically ranging from 2 to 50 amino acids. The BCRs bind to variable sized proteins through the epitopes of protein, consisting of 5–20 amino acids, and in general, TCR-binding peptides are 8–20 amino acids in MHC-restricted cases, but longer peptides can bind to TCRs [51]. The adaptive immune cells against the proteins/peptides are thought to have evolved primarily to respond to internal antigens, since control of these bioactive antigens is critical for cell protection.

As for innate immune components, their presence in pathology also reflects an ongoing immune attempt to contain and neutralize the toxic substances. The lack of specific immune components against substances is responsible for certain types of chronic inflammatory conditions. For instance, there are some diseases, in which immune cells and immune proteins such as immunoglobulins and complements are not observed including idiopathic nephrotic syndrome, Reye syndrome, amyloidosis, and chronic neurovegetative diseases such as prion diseases and Alzheimer’s disease. In these cases, only amyloid proteins/peptides are deposited in the early pathological lesions. The PHS hypothesis suggests that prions and amyloid proteins are not etiological substances but the immune effectors against smaller toxic substances targeting nerve cells or other organ cells. The chronic activation of these non-specific effectors—such as prions or abnormal immune proteins—is responsible for disease progression in the absence of specific immune proteins that control initial etiological substances and/or those derived from injured target cells [12].

Each cell contains multiple organelles (e.g., nucleus, mitochondria, lysosomes, ribosomes, and proteasomes) as well as diverse proteins, peptides, and epigenetic elements such as microRNAs. Although all cells share the same genome, the expression and utilization of proteins, peptides, and other biomolecules vary significantly, and receptor profiles also differ depending on cell type in various organ tissues. Intercellular communication predominantly occurs via the substance–receptor–signal pathways [6]. When toxic intracellular substances are released from certain organ cells and bind to affinitive receptors on neighboring or distant target cells, immune components are inevitably recruited in response. Moreover, intracellular infections caused by microbiota or external pathogens can lead to the production of more diverse proinflammatory substances.

The existence of localized pathogenic foci underlies the pathogenesis of a wide range of infection-related immune-mediated conditions, including autoimmune and chronic inflammatory diseases, as well as classical infectious diseases. This concept dates back to the era of old medicine, as Hippocrates described a patient with rheumatism whose arthritis was cured by the extraction of a tooth, the presumed focus of arthritis [52]. In this context, a focus can be defined as a localized lesion in extracellular pathogen infections, or as a localized legion(s) consisting of initially infected cells in intracellular infections. While the focus can often be readily identified in extracellular infections such as cellulitis, osteomyelitis, appendicitis, and periodontitis, it remains challenging to localize the focus in systemic intracellular infections—such as those caused by viruses, *Chlamydia*, *Rickettsia*, or intracellular bacteria like *Salmonella typhi* and *Brucella* species—even with advanced diagnostic techniques. The inflammation-inducing substances from the focus can spread to localized and remote areas. For example, localized infections such as periodontitis or sinusitis can trigger systemic inflammation, affecting both nearby and remote organs [53,54]. Moreover, injured host cells at the focus also release intracellular contents, with or without pathogen-derived materials, thereby exacerbating inflammatory responses elsewhere in the body. Accordingly, extracellular and intracellular infections and autoimmune diseases can induce secondary organ specific diseases including acute or chronic arthropathies, cardiovascular disorders, pulmonary involvements, and other organ diseases. Another example is bilateral orchitis occurring after parotid gland swelling in mumps, suggesting that inflammatory substances released from damaged parotid gland cells affect both testicular cells [55]. It is thus plausible that intracellular content released from certain types of injured cells may induce multiple organ involvement as observed in SLE. Similarly, substances from initially infected cells in diseases such as COVID-19, influenza, and *Mycoplasma pneumoniae* infection affect diverse target organs, leading to extrapulmonary manifestations such as skin rashes, arthritis, encephalopathy, and other organ involvements [8,9,10]. In therapeutic perspective, if a chronic disease has the focus, controlling the focus could reduce morbidity and prevent progression of the diseases.

### 5.2. Target Cells and Receptor-Mediated Inflammation

If target cells lack receptors for specific etiological substances, inflammatory responses are unlikely to occur. Moreover, if immune components capable of recognizing these substances are lacking, the resulting inflammation tends to be less intense. For example, in the case of snake venoms, a venom is composed of various proteins, peptides, and biochemicals as an external natural toxin, and the affected target cells vary depending on the venom’s components. Some patients do not respond to anti-venom antibodies because toxins are peptides or non-protein substances [56,57]. Within a host cell in viral infections, the generation of a single virion results in production of more than 1000 times the number of viral byproducts, including capsid proteins and viral nucleic acids. In addition, numerous host’s immune components such as interferons and other antiviral proteins are produced [9]. Not only respiratory viruses, including influenza viruses and coronaviruses, but also non-infectious conditions, including blunt chest trauma, multiple traumata, severe burns, smoke inhalation, gastric content aspiration, acute pancreatitis, and amniotic fluid embolism can cause pneumonia and acute respiratory distress syndrome (ARDS) with remarkably similar clinical and histopathological features [11]. Patients with acute and chronic infection-related immune-mediated conditions such as KD, MIS-C, RA, and SLE, exhibit variable target organ involvement, often differing from one individual to another. These findings suggest that the receptors of target cells that respond to intracellular substances and/or pathogen-derived substances are critical rather than the receptors for portal of pathogen entry. There are species-specific microbial strains in each animal species, including influenza viruses, coronaviruses and mycoplasmas. T cell deficient animals infected with these pathogens show less pulmonary inflammation compared to immune-competent controls [8,9]. The findings suggest that inflammation-inducing substances are pathogenic peptides derived from initially infected cells, and respiratory endothelial or epithelial cells serve as the primary target cells [8,9,10,11].

### 5.3. Phenotype, Pathology, and Pathogenesis of Diseases

Clinical symptoms result from the interaction between toxic substances (or etiological substances) and immune components attempting to neutralize them. Accordingly, both the quantity and biochemical nature (or virulence) of these substances critically influence disease severity and phenotypic expression. In intracellular infections, when more cells are initially infected and injured, more etiological substances are produced, which is related to the severity of the disease. Besides the load of etiological substances, the phenotypes of an intracellular infectious disease are determined by target cells of etiological substances. Theoretically, the composition of inflammatory substances within infected cells differs depending on whether the same pathogen infects distinct cell types, or different pathogens infect the same cell types. Etiological substances include not only immunogenic proteins and peptides, which are regulated by the adaptive immune system, but also smaller substances including elements, monoamines, chemicals, neuropeptides, and biochemicals. These smaller molecules typically do not elicit antibody production, implying the involvement of additional immune proteins or unidentified immune components for their clearance.

Virtually any host cell can bind to toxic substances, and immune responses to the substances appear in pathological tissues. Early-stage histopathological findings often reflect the immune system’s attempt to control etiological agents, providing valuable clues to their identity [7,12]. The immune cells and immune proteins such as immunoglobulins and compliments observed in histopathology are gathered according to the need of the host for cell protection. For example, if eosinophils are deposited, toxic substances are controlled by eosinophils and/or eosinophil-associated immune responses, and if IgEs are deposited, etiological substances are controlled by IgE and/or IgE-associated inflammatory responses for cell protection.

In autoimmune diseases, autoantibodies and self-reactive T cell clones are not etiological substances or causal effectors that attack self-cells via recognition of antigens expressed on/in cells. Instead, they function as critical responders against toxic substances released from injured self-cells for protection of emerging cell injury. Tissue injury in autoimmune diseases is primarily driven by the sustained activity of non-specific adaptive immune cells, arising from the absence or insufficiency of specific T or B cell clones capable of effectively regulating pathogenic peptides and proteins. When injured cells discharge their intracellular contents into the circulation and then pathogenic and bioactive peptides and proteins bind to receptors on target cells, T cells and antibodies are consistently detected within pathological lesions.

### 5.4. Integrated Immune Response and Lack of Specific Immune Effectors

The immune system under the PHS is responsible for cell protection from various insults. For this, all immune components communicate with each other through mechanisms involved in major histocompatibility complexes (MHCs), cytokine networks, and possibly mast cell-associated networks. In acute immune-mediated diseases, target cells are injured by cytokine imbalances such as cytokine storm caused by non-specific T cells and/or B cells (antibodies), and other immune components, which act early stage of the diseases against large loads of etiological substances. In contrast, chronic immune-mediated diseases reflect a failure of T cells, B cells, or other immune components to fully eliminate pathogenic substances derived from damaged target cells. These unresolved substances can exert pathogenic effects on adjacent or distant cells in organs. Also, it can be inferred that non-specific immune components instead of specific ones control etiological substances of the disease in part in chronic inflammatory diseases and autoimmune diseases [5,6].

The composition and functional capacity of immune components are largely determined by the host’s genetic architecture. Therefore, T cells and B cells’ specific immune response to pathogenic protein/peptide antigens may depend on the genes that constitute TCR and BCR. Similarly, the expression and function of other immune components—including complement proteins, prions, and amyloidogenic proteins that may target small toxic molecules—are also genetically regulated [12]. Moreover, the function of the immune system changes throughout life, and therefore the severity and prognosis between children and adults differ in infectious and infection-related immune-mediated diseases [5,6].

### 5.5. PHS Perspective on Cancer and Genetic Disorders

Cancer and genetic diseases begin with a protein deficiency (or single gene defect) within the cells or in the host, respectively. To overcome the defective protein(s), compensatory mechanisms of the intracellular PHS in cancers and systemic PHS in genetic diseases act and significantly influence on the development, clinical manifestation and the overall prognosis of such diseases. For a more comprehensive discussion of these concepts, readers are referred to previously published studies [7,12,13]. A summary of immune components associated with various etiological substances is provided in Table 1.

## 6. Antigens, Allergens, and Etiological Substances

### 6.1. Antigens: Proteins/Peptides

In the classical sense, antigens are defined as substances capable of inducing antibody production, typically of pathogen-origin. Most antigens are proteins. During infection, B cell clones that express BCRs capable of recognizing a specific antigen initially produce IgM antibodies. Subsequently, class-switched clones that produce IgG, IgA, and IgE are generated [58]. These antibodies can react to free antigens in the blood, and antibody-bounded antigens lose the biological activity of the antigen, including their toxic effects, a process referred to as detoxication or neutralization. The antibodies react also to antigens presented on surface of pathogens or target cells. Therapeutically, anti-toxin antibodies, monoclonal antibodies acting on proinflammatory cytokines or cytokine receptors have been successfully applied in many diseases, including allergic diseases [59]. IgG, IgA and IgE isotype antibodies differ in tissue distribution, serum concentration, and biological functions in vivo. The reason for existence of isotype antibodies against a specific antigen remains unknown. It can be suggested that the various isotype clones effectively control critical antigens according to their route of entry and tissue localization, and that IgE isotypes are important in recognizing external antigens that are associated with mast cells distributed in interface areas.

T cells become activated upon recognizing peptide fragments of protein antigens that are processed and presented by antigen presenting cells (APCs). Although different subsets of T cells perform distinct functions, all T cell clones are activated through peptides linked between their TCRs and MHCs, suggesting that peptides are an essential part in MHC-restricted specific immune responses. The peptides have variable sizes to bind TCRs, and smaller peptides such as neuropeptides or larger peptides as small proteins that cannot bind to TCRs, cannot activate and proliferate T cell clones. Similarly to antibodies, the precise regulation of bioactive peptides may be critical for host cell protection. There are cross-reactive clones in cytotoxic and helper T cells, which possess TCRs recognizing the same epitopes of peptide antigens, suggesting that various T cell subsets are directed toward identical peptide antigens. It is suggested that T cells control pathogenic peptides through production of cytokines (and possibly peptides) and materials within the cells [6]. On the other hand, the subsets of T cells react with free pathogenic peptides circulating in the blood as well as pathogenic peptides bound to various receptors, including MHCs, on both self and foreign cells. This can be regarded as the MHC-non-restricted immune response. In such cases, non-specific or cross-reactive T cells initiate the response against excessive load of toxic peptide antigens, which can subsequently lead to cytokine storm, ultimately resulting in host tissue cell injury.

B cells generally cannot produce antibodies against the peptides controlled by T cells, and small non-protein molecules such as neuropeptides, chemicals, and biochemicals. Some non-protein materials, called haptens, can bind proteins in vitro and then produce antibodies in vivo, and various carrier proteins are used in vitro for vaccine development [60], suggesting that many unidentified haptens and carrier proteins can be to occur naturally in vivo. Anti-double strained DNA antibodies are a landscape of diagnosis of SLE, but results of detection methods somewhat vary depending on the used antigens in assays, such as synthetic DNAs or other forms of the antigen [61]. The anti-double strained DNA antibodies in vivo may be produced against the fragmented double strained DNAs attached to proteins such as histones [62]. Similarly, anti-citrulline antibodies, which serve as a specific biomarker for RA, recognize the proteins containing citrulline [63]. Some natural antibodies can bind to non-protein biochemicals like vitamin D in vitro, but such responses in vivo may involve haptenated protein antigens [64].

### 6.2. Allergens

Allergens are defined as substances that cause allergic reactions. It has been proposed that such reactions serve as an important defense mechanism, protecting the host from environmental or external toxic materials such as venoms, natural toxins, and arthropod bites [65,66]. These natural toxins are composed of complex materials as well as other allergens including pollen, house dust mites, and foods. Importantly, the allergenic reaction is not necessarily triggered by complex allergens as a whole, just as viruses and bacteria themselves do not function as toxins in the pathogenesis of infectious diseases. Also, certain constituents of external or internal pathogens, including viruses, bacteria, and fungi, induce production of IgE [67], supporting that pathogen infections can play a possible role in allergic diseases. Complex allergens often contain diverse molecular constituents including proteins, peptides, biochemicals, chemicals, and metals like nickel and chromium. For instance, food is not only composed of nutrients such as proteins, carbohydrates, and lipids, but also includes potentially pathogenic proteins detectable by IgG (including IgG4) and IgE. Moreover, foods contain bioactive peptides, neuropeptides, monoamines, minerals, vitamins, fatty acids, nucleic acid fragments, food additives and other unidentified substances, and food allergies became one of main topic in allergies [68].

The fractional analysis of allergens has been tried to identify their detailed components, but it remains challenging to isolate and characterize specific allergenic substances. Certain protein fractions of an allergen can bind small materials, called ligands, such as lipids and/or lipid derivatives, and the protein-ligand substances potentially act as etiological substances on allergic reactions [69]. Allergen extracts used in the Prick skin test are themselves complex and contain substances that activate mast cells in the dermis. Similarly, the antigens used in diagnostic kits such as RAST and CAP for detecting sensitized IgE contain pathogenic proteins, including those bound haptens such as lipids, as a complex material [70]. While foods are considered exogenous allergens, they are digested in gastrointestinal tracts, and some components are absorbed into the blood. Thus, food acts as an internal allergen. Because the immune system cannot produce antibodies against small substances without haptens in a steady state, diagnostic methods for allergic diseases, especially food-induced disorders, have had some limitations [68,71]. Heathy nonatopic persons can have specific IgE antibodies on allergens, and some atopic patients can show no allergic reactions in the provocation test. These phenomena are seen in autoantibodies in autoimmune diseases, suggesting that not only protein antigens but also other etiological substances are related to the diseases with corresponding immune responses to them.

People living in modern society are exposed to more environmental chemicals such as exhaust fumes, pollution, and food additives than those in the past. Many studies have suggested that the increased incidence of diseases, including allergic or autoimmune diseases in the recent era are related to microbe-rich environments and increased chemical exposures [72,73]. Chemicals, elements, and biochemicals as the constituents of allergens can be in the air, foods, pathogens including fungi, and working environment [74]. Moreover, these substances can be present on the surface of complex allergens such as pollens, house dust mites, animal dander, and cigarette smoke, and potentially contribute to mast cell activation and mast cell-mediated inflammatory responses. Also, some chemicals, including pharmaceutical agents, can bind to certain proteins as haptenated proteins in vivo, thus behaving like protein antigens capable of eliciting adaptive immune responses. Drugs such as amino salicylic acids (aspirin), antibiotics, and non-steroid anti-inflammatory drugs bind to target cells and are regulated by catabolic mechanisms involving carrier proteins and other enzymes [75]. Within the PHS hypothesis, some proteins involving pharmacokinetic catabolism are regarded as an immune effector, since excessive drug exposure can lead to severe or fatal outcomes.

In certain patients, psychological factors—often shaped by sociocultural or educational influences such as the belief that “this substance is harmful”—contribute to the development of allergic or allergy-like symptoms through incompletely understood neuropsychological mechanisms. For example, emotional upset, exposure to cigarette smoke, certain perfumes, or animal odors can trigger asthmatic episodes. Interestingly, cigarette smoke was historically used as therapeutic measure for severe asthma for several decades [76]. The psychological placebo effect may influence symptom perception or clinical outcomes in certain patients with asthma (please see Section 7.4. Mast cells for further discussion).

### 6.3. Pathogenic Substances Within Cells

Proteins, as products of encoded genes, play critical roles in organisms and individual cells. Protein homeostasis within cells, called proteostasis, is an important concept in cell biology [77]. The protein levels in the blood, including those of albumin and immunoglobulins, are also tightly regulated, although fluctuations occur in pathological conditions [78]. These observations suggest that maintaining a balanced proteostatic environment is essential for organismal survival and cellular function. Disruptions in proteostasis have been implicated in various diseases including neurodegenerative disorders and cancer [79,80].

Proteasomes—present in both the nucleus and cytoplasm of all nucleated cells from yeasts to humans—are key players in proteostasis. Their distribution and type vary by cell type in organ tissues, and their main function is the degradation of modified or misfolded proteins within cells [77]. In addition to their degrative function, proteasomes also generate a variety of intracellular peptides, some of which exhibit biological activity within cells including those in the CNS [81]. The comparison of peptidome has revealed similar protein/peptide profiles between yeast and mammalian cells, suggesting that intracellular peptides have specific and conserved biological functions [82]. Thousands of naturally occurring peptides have been identified across vertebrates, plants, insects, fungi, and bacteria, differing in origin, abundance, and functions [83]. Moreover, studies have shown significant sequence homology between human and bacterial proteins, implying that similar peptides are produced and utilized across species [84]. These findings support the hypothesis that bioactive peptides are ubiquitous and serve as essential regulators of cellular viability and homeostasis. Now proteasome inhibitors are used in multiple myeloma and other immune-mediated diseases and act to reduce intracellular peptide levels [85]. Their therapeutic effects may arise from blocking the production of essential peptides for cell survival in multiple myeloma cells and other cells. Notably, cellular injury caused by infection, trauma, or cytokine-mediated injury leads to the release of peptides, including pathogenic ones, into the systemic circulation. The bioactive and signaling peptides bind to high-affinity receptors on target cells of various organs and initiates T cell-associated inflammations, as proposed in the PHS hypothesis.

Not only proteins/peptides but also small bioactive molecules, including elements (e.g., oxygen, nitroxide, potassium), monoamines (e.g., acetylcholine, serotonin, dopamine), small peptides (e.g., neuropeptides, peptide hormones), and various biochemicals (e.g., vitamins, fatty acids), should also be tightly regulated to sustain systemic and cellular homeostasis. Thus, it is possible that the activation of eosinophils, mast cells, and basophils is associated with the endogenous metabolites released from injured cells.

## 7. Immune Components in Allergy Under the PHS Hypothesis

Patients with allergic diseases show an increased levels of biomarkers indicative of activated Th2 cells, eosinophils, mast cells, and IgE in both the blood and pathological lesions, together with increased activity of various components of the adaptive and innate immune systems [86]. In this section, we briefly discuss these findings, with a particular focus on IgE, eosinophils, and mast cells.

### 7.1. T Cells and MHCs

T cells, including helper T cell (CD4+) cells and cytotoxic T cells (CD8+), are involved not only in infectious diseases but also in wide range of non-infectious diseases such as allergic diseases, autoimmune diseases, tissue regeneration (or wound healing), organ transplantation, intoxication, and cancer. This broad involvement suggests that their functions and activation mechanisms are conserved across diverse pathological contexts. In MHC-restricted immune response, peptide antigens presented by (i.e., bound in) MHC molecules on APCs are recognized by TCRs, leading to the generation of antigen-specific T cell clones. T cell activation is further tightly regulated by co-stimulatory and inhibitory receptors including CD28, CTLA-4, and PD-1. It is accepted that complex protein antigens taken up by APCs serve as sources of peptide antigens for T cell activation. These peptides possess pathogenic potential and must be tightly regulated to prevent emerging damage to host cells. Thus, MHCs function as peptide-binding receptors that present self-peptides, including those essential for the organisms, and serve as self-identity markers reflecting their critical role in the MHC-restricted immune response. In case of organ transplantation, B cells recognize foreign MHC molecules as pathogenic proteins, while T cells target foreign peptides presented on MHCs or other surface receptors of donor cells within the PHS hypothesis [6]. Also, T cells are rapidly activated by free peptides, superantigens, mitogens, drugs, antibodies targeting TCRs, in both in vitro and in vivo, independent of MHCs [87,88]. Thus, T cell responses are categorized into 2 pathways; the MHC-restricted pathways, which involve specific recognition of peptide within MHCs, and the non-MHC-restricted pathways, which are more broadly reactive and temporally distinct as previously discussed. The composition and activity of T cell subsets, including Th1, Th2, Th17, Treg and cytotoxic T cells, vary even among patients with the same disease and change during disease process for recovery of immune homeostasis [89]. These variations are likely influenced by the origin and nature of pathogenic peptides and corresponding T cell subsets in disordered conditions.

### 7.2. B Cells and IgEs

B cells produce immunoglobulins, which play a key role in neutralizing pathogenic proteins. Total serum IgG level maintains relatively stable in mature adults, although interindividual variation exists. During systemic infections or infection-related immune-mediated diseases, total IgG levels typically begin to rise at the early convalescent stage and return to baseline after convalescence, suggesting that IgG works for resolution of the diseases [10,78]. Structurally, IgE shares similarity with IgG and IgA, but significantly lower level in serum. The adaptive immune system matures with age, and infants generally exhibit low levels of IgM, IgG (except maternal origin), and IgA, with minimal detectable IgE. This can be explained that there is little exposure to both endogenous protein antigens and exogenous antigenic stimuli such as allergens. Besides allergic diseases, elevated serum IgE levels have been overserved in chronic immune-related diseases including idiopathic nephrotic syndrome, hyper-IgE syndrome, and some immunodeficiencies, often exceeding the levels typically seen in allergic diseases [90]. Additionally, IgE can be produced in response to viral and bacterial infections and detected as autoantibodies in autoimmune diseases [67,91,92]. Selective IgE deficiency is associated with increased susceptibility to more severe allergic conditions, autoimmune disorders, and malignancies [93,94], suggesting that IgE play a broader protective role in the immune system. Notably, studies have reported that IgE level in nasal secretions increases following nasal allergen challenges in patients with both allergic and non-allergic rhinitis [95], suggesting that IgE concentration in local lesions is higher than those in the blood. If IgE production results from class switching of a single B cell clone against a pathogenic protein antigen, the very low concentration in the blood compared to IgG and IgA should be explained. One plausible explanation is that IgE primarily targets free, exogenous pathogenic proteins, such as those in environmental toxins or allergens. Consequently, IgE-producing B cell clones are likely to reside at interface or barrier sites such as the intestines, airways, and skin, like the distribution of mast cells, where they respond to external toxic proteins rather than those circulating in the blood. Thus, IgE has relatively limited involvement in controlling self-cell derived protein antigens. Specific IgEs in interface regions react immediately to sensitized antigens that invade from the outside and induce detoxication of the antigens. Whereas non-specific IgEs act against structurally similar novel antigens through cross-reactivity with IgE-associated inflammation. Like IgG, IgE can bind to antigens bound on cell surface receptors—particularly on mast cells—and activate mast cells at barrier sites. Thus, it is assumed that increased total IgE levels arise when prolonged exposure to external antigens (and occasionally internal antigens) stimulates B cell clones at localized lesions, leading to release of IgEs into the bloodstream under certain physiological or pathological mechanisms.

IgG4-related diseases are characterized by elevated levels of IgG4, a subclass of IgG known for its structural and functional heterogeneity [96]. Specific IgG4 antibodies against food antigens have been observed in non-IgE-mediated food adverse reactions, and elevated allergen specific IgG4 levels have also been observed in patients with IgE-mediated allergic diseases [97]. According to the PHS hypothesis, increased IgG4 is not the cause of IgG4-related diseases, but rather a consequence of immune reactions aimed at neutralizing pathogenic proteins as well as the functions of other immunoglobulins and their subclasses. Thus, IgG4 may modulate immune responses caused by pathogenic proteins that have distinct molecular forms and bioactivities and encountered in food-related and other IgG4-related diseases.

### 7.3. Innate Immune Cells and Eosinophils

Immune cells in the innate immune system include monocytes/macrophages, NK cells, granulocytes (neutrophils, eosinophils and basophils) and tissue macrophages and mast cells. The main function of these cells is defense of the host, primarily through the protection of cells. During mammalian embryogenesis, animals lacking T or B cells are born with structurally and functionally intact organs. In contrast, the absence of innate immune cells, especially those of the macrophage lineage, result in nonviable development. This observation suggests that the innate immune cells involved in embryonal development have diverse integrative functions beyond immune defense, in contrast to T cells, which do not appear to contribute to integrative functions. Recent studies have expanded our understanding of granulocyte functions. Neutrophil, eosinophil, basophil, and mast cells contain specific granule contents and lipid bodies and express various receptors, including chemoattractant, cytokine, growth factor, adhesion, Fc, and pattern-recognition receptors, as well as receptors for biochemical mediators such as lipid compounds [98,99,100]. These multiple cell-derived mediators and receptors across innate immune cells may be needed to communicate with each other for cell protection. More importantly, some intracellular content, including performed mediators in granules, can be used against internal and external toxic insults. The granules of these cells contain numerous proteins and biochemicals, and some of them can be cytotoxic when released inappropriately. In healthy individuals, eosinophils and basophils are typically present at low proportions relative to neutrophils and lymphocytes in the blood. Mast cells and basophils are known to have similar immunological roles and mechanisms, particularly in allergic inflammation [101]. Notably, mast cells are abundantly distributed across nearly all tissues that interface with the external environment, whereas basophils are relatively rare in circulation, akin to the low serum levels of IgE relative to IgG. Both mast cells and basophils express high-affinity receptors for IgE and mediate immune responses through IgE-binding. As described earlier, IgE mainly recognizes and cross-reacts with toxic protein antigens that originate externally. Therefore, within the PHS hypothesis, mast cells have affinitive receptors for external pathogenic proteins, and specific or cross-reactive IgEs target the antigens bound to receptors expressed on the mast cells/basophils and trigger immune responses, rather than known action of the antigens which induce crosslinking of antigen-specific IgE bound to high affinity receptors for IgE on the mast cells’ surface.

Neutrophils function primarily as phagocytes, eliminating pathogens such as viruses and bacteria during infection. Typically, neutrophils are the first immune cells to arrive at sites of cell injury or danger, followed sequentially by lymphocytes. For example, in pulmonary lesions in animal models of influenza or mycoplasma pneumonia, neutrophils appear as the first line of defense followed by lymphocytes [8]. A similar temporal pattern is observed during wound healing processes following trauma and in certain asthma models [102,103]. It is plausible that upon cell injury, neutrophils initially clear larger apoptotic or necrotic bodies released from damaged cells, while small substances are addressed by antibodies, T cells, and other immune components.

Basophils are thought to have functional similarities to mast cells, including capacity to release intracellular substances such as granule proteins and biochemicals during allergic reactions [100]. Notably, IgE-mediated hypersensitivity reactions to peanut components have been shown to involve basophils rather than mast cells [104]. This led to the clinical application of basophil activation tests for the diagnosis of allergic diseases including food allergies [105]. Nonetheless, it remains unclear whether the etiological substances that activate mast cells also activate basophils. It is plausible that basophils as circulating mast cells are activated when they are exposed to immunologically active substances released from local lesions into the blood.

Historically, parasites such as roundworms and insects like scabies commonly coexisted with human hosts, inhabiting the intestines and skin. At various stages in their life cycles, including egg and larval forms, these organisms can enter host tissues on occasion. The connection between eosinophils and parasite infections was recognized early in development of immunology [106]. In parasitic infestation, including larva migrans and Loeffler’s syndrome, the proportion of eosinophils in the blood increase more than 50% of the total white blood cells [107]. In addition, eosinophils are the predominant cells found in mucus in some type of fibrinous or plastic bronchitis that cause mucus plug syndrome [108]. Eosinophil-dominant infiltration is also observed in diseases like eosinophilic granulomatosis and eosinophilic gastroenteropathy [109]. Eosinophilic esophagitis has been reported during oral immunotherapy [110], suggesting that certain allergen extracts contain etiologic substances eliciting eosinophil-inducing inflammation. In addition, blood eosinophil levels vary across ethnic groups, and individuals in parasite prevalent areas show a lower prevalence of allergic diseases. These findings suggest that eosinophils are involved in immune response to parasite invasion and recruited in large numbers in localized lesions and blood. It is possible that eosinophils recognize and respond to toxic bioproducts such as unique biochemicals including haptenic protein/peptides secreted by living parasites in host tissues. Similar molecular substances in environmental allergens or certain food components induce eosinophil-associated inflammation. Since eosinophils can be activated through cross-reactivity, excessive eosinophil-associated inflammation affects living parasites and host’s tissue cells, and injured host-cells can initiate adaptive immune responses. Although the specific external substances that trigger eosinophil activation are not fully characterized, they likely have distinct bioactive properties and are not typically found in the host cells. It is thought that certain intracellular factors within eosinophils control these external toxic substances.

### 7.4. Mast Cells, the Mast Cell-Associated Network, and Psychological System

Mast cells, first identified in the 19th century, are distributed throughout all vascularized tissues and are predominantly located at environmental interfaces, including the skin, airways, and gastrointestinal tract. Notably, they are also found within the central nervous system (CNS) [111]. Mast cells are activated by insults including allergens, infectious agents, and endogenous stressors, however precise etiological substances for mast cell activation remain incompletely defined. While mast cells exhibit tissue-specific heterogeneity, their primary role is considered to protect the host cells in various tissues as components of the innate immune system. Upon activation, mast cells release a diverse spectrum of bioactive proteins and biochemicals, including proteases (e.g., tryptase, chymase), cytokines, growth factors, histamine, heparin, eicosanoids (e.g., prostaglandins and leukotrienes), and neuropeptides. These materials may be associated with resolution of inflammation and action of adaptive immune cells [112]. Mast cells have diverse immune functions, including host defense against pathogens and wound healing, reflecting their traditional role as one of the most ancient components of the immune system [113,114]. Although mast cell and IgE-mediated immediate immune responses such as anaphylaxis are well described, there are also non-IgE associated reactions in allergic diseases. It is believed that mast cells contribute to non-IgE associated immune responses in both allergic and non-allergic diseases [115,116].

In the CNS, an immunologically privileged site, the presence of adaptive immune components including T and B cells as well as complement proteins, is limited under homeostatic conditions. Only mast cells, glial cells as the form of tissue macrophages, and limited immune proteins in cerebrospinal fluid are present. Effective communication between neurons and these immune-related cells is essential for maintaining CNS homeostasis. CNS regulates voluntary behaviors, involuntary physiological functions (e.g., cardiac, gastrointestinal, and respiratory activity), and psychological or emotional insults such as anxiety, depression, and post-traumatic syndrome. Although consciousness and emotion arise within the brain, they may be a part of essential aspects of the biosystem (living organism) and are difficult to define in purely material terms [12,13]. Psychological or emotional responses are influenced by networks of complex CNS cells, and severity of depression, anorexia, and sleep disturbance are highly different in individuals and can function analogously to involuntary CNS-mediated processes in certain events. The networks of CNS cells for regulation of psychological phenomena are called the “psychological system” in this article.

It is now well established that neuroactive substances, including serotonin, monoamine precursors, and neuropeptides are produced in the gastrointestinal tract supporting the ‘microbiota-gut–brain-axis’ (or ‘gut–brain axis’) hypothesis. According to this framework, brain health, including stability of psychological and psychiatric function, is closely related to intestinal environment, particularly in dysbiosis or infection [117]. Dysregulated immune activation within CNS has been implicated in the pathophysiology of bowel dysfunctions such as dyspepsia and irritable bowel syndrome [118]. Psychological insults, mainly acute and chronic stress, have also been associated with psychosomatic disorders including depression, gastrointestinal troubles, circulatory system troubles, exacerbation of allergic diseases and possibly autoimmune diseases [119,120]. Chronic stress modulates various immune functions, suggesting that the CNS immune system influences systemic immune systems through communication pathways. Also, exposure to allergens in sensitized animals or humans showed behavioral alteration such as anxiety-like signs or worsening of psychological diseases [121,122], and perceived stress in individuals was reported to be associated with allergic flares [123]. These findings suggest that the psychological system in the CNS is related to systemic immune functions including allergic responses.

Although the precise role of mast cells in the CNS remains to be fully elucidated, toxic neurotropic substances originating from the intestines or other external sources activate mast cells in CNS, particularly if these substances can cross the blood–brain barrier. The peripheral nervous system, which connects the CNS to distal tissues, extends to the peripheral tissues where mast cells are abundantly distributed. It serves as a bidirectional pathway for afferent and efferent signals between local inflammatory sites and the CNS in disordered conditions including food allergy and pain syndromes [124,125]. Thus, it is possible that not only sensory information—such as pain or pressure—but also inflammatory cues, including mast cell activation status, are transmitted to the CNS. Reciprocally, activated mast cells in the CNS influence peripheral mast cell responses through efferent mechanisms. This bidirectional communication is referred to as the “mast cell-associated network” in this article. Mast cell activation syndrome (MCAS) has recently been recognized. Although it shares features with systemic mastocytosis, MCAS is more prevalent and is characterized by minimal or no evidence of clonal mast cell expansion. Patients with MCAS exhibit a broad range of acute and chronic conditions affecting multiple systems, often involving allergic and inflammatory symptoms, as well as various neurological and psychiatric manifestations [126]. The substances causing mast cell activation or mediators from activated mast cells in the peripheral lesions can circulate systemically and influence mast cell-related immune responses in the CNS. Conversely, mast cells activated in the CNS manifesting neurological or psychiatric symptoms can induce other symptoms of MACS such as allergic reactions through efferent pathways. Researchers have proposed that mast cells in the CNS are involved in various diseases, including neurodegenerative and autoimmune diseases, as well as psychological and psychiatric disorders [127,128,129]. Notable psychological phenomena in the psychological system include the placebo effect, phantom phenomena (e.g., phantom limb or eye pain), classical conditioning as seen in Pavlov’s dog experiment, and the secondary gain from unintended behaviors highlight the complexity of the psychological system [130,131,132,133]. Thus, psychological imprints established through repeated emotional or physical stimuli affect individual behaviors and possibly immune responses that are associated with the mast cell-associated network.

The precise identify of etiological or toxic substances controlled by mast cells and mast cell-associated immune responses remain unknown. They may activate distinct mast cell subsets in target organ tissues and vary depending on the pathological conditions. The substances are very small of either external- or internal-origin including chemicals, biochemicals, monoamine-derivatives or metabolites, and neuropeptides, particularly in the context of psychological system. The immune effectors responsible for controlling these substances are likely contained within mast cells themselves. Thus, inflammatory mediators derived from activated mast cells, together with these immune effectors, play key roles in eliciting the clinical manifestations of allergic disease.

## 8. Pathologic Findings in Allergic Diseases

Inflammation is a phenomenon in which immune components, including immune cells and immune proteins such as immunoglobulins and complements, appear histologically in affected lesions. Accordingly, allergic diseases such as asthma and AD demonstrate various immune cells and immune proteins in pathologic lesions. Ciliated epithelial cells and goblet cell-derived mucus act as a first line of defense against pathogens and allergens from the outside. The respiratory tract plays a role as a passway for clearing damaged cells, infiltrating immune cells, and byproducts generated at local inflammatory lesions such as asthma, pneumonia, and bronchiolitis. The study of the bronchial alveolar lavage (BAL) is useful for diagnosing and managing pulmonary diseases. Species-specific bacterial or viral strains such as influenza viruses, corona viruses, and *Mycoplasma* species can present (i.e., colonization, not infection) in upper or rarely low respiratory tracts of some healthy people, especially in children as carriers or reservoirs, especially during epidemic periods. Therefore, microbial detection in BALs cannot be confirmative for causative substances eliciting asthma or pneumonia [10,11].

BAL samples from asthma patients frequently contain various immune cells such as neutrophils, eosinophils, and T cells as well as immune mediators, including immunoglobulins, Th1- and Th2-associated cytokines, mast cell-derived mediators (e.g., histamine, leukotrienes), and eosinophil-derived materials. Besides allergens, asthma is exacerbated by various stimuli including viral infection, exercise, and acute emotional stress. During allergen-induced asthma attack, innate immune cells such as eosinophils and neutrophils are predominantly observed in the early stage, and lymphocytes become more prominent in the later stages [134]. Immune cells involved in lung lesions, including asthma and pneumonia, are discharged from the localized inflammatory sites via bronchial branches. The immune components in BAL in asthma vary depending on the triggering factors such as allergens and pathogens, and the timing of sampling relative to symptom onset. Asthma endotypes have been classified based on the immune cell profiles observed in mucus (or sputum) or BAL: in general, type 2 (T2) high (eosinophilic) or T2-low (neutrophilic or pauci-granulocytic) asthma [135,136]. Since both immune cells and immune proteins seen in mucus or BAL are discharged from mainly localized lesions, blood levels of total eosinophils and total IgE do not often correlate with clinical severity in T2-high asthma. As previously discussed, neutrophils and lymphocytes in pathological lesions are associated with cell injury, thus the pathophysiology of the T2-low asthma may be more directly linked to cell injury of the respiratory tracts and/or the lungs. Biomarkers identified in mucus or BAL reflect the inflammatory status and represent the dynamic interaction between etiological agents, including those derived from allergens or pathogens, and the host immune response in the localized lesions. In allergen-induced asthma, early-phase immune activity may be dominated by eosinophils and mast cells, whereas in infection-driven asthma, neutrophils and T cells may predominate in the localized lesions as the focuses. After this local immune activation, inflammation-inducing substances derived from hyperactive immune cells and/or those derived from damaged cells propagate through the systemic circulation, and the substances reach target cells in both lung tissues and contribute to bilateral bronchoconstriction and inflammation (please see 10. Pathophysiology section).

## 9. Microbiota in Allergic and Autoimmune Diseases

Microorganisms, including viruses, bacteria, and fungi, are ubiquitous across the Earth, with strains identified even in extreme environments and artificial structures such as buildings [137,138]. However, only a subset of these microbes interacts with or affect human hosts. The microbial community within animals, that is, microbiota, has coevolved in a symbiotic relationship with their mammalian hosts including the development of immune systems [139]. The composition of human microbiota varies with age and differs in individuals, ethnic groups, and populations [140]. Environmental factors, including diet, antibiotic exposure, and potentially probiotic use, influence the diversity and composition of these microbial communities [141]. It is well established that the emergence of a new disease within populations is often linked to environmental changes. Among environmental factors, alternations in the human microbiota are of relevance to the pathophysiology of diseases. The ‘hygiene hypothesis’ and related theories suggest that individuals in modern societies are exposed to fewer infectious agents during early life, resulting in insufficient immune system training. This underexposure leads the immune system to react inappropriately to harmless antigens, such as allergens [142]. The ‘microflora hypothesis’ suggests that strains in microbiota have evolved along with changing environment and contribute to increased cases of allergic diseases [143]. Recently, the ‘epithelial barrier hypothesis’ propose that a rise in agents detrimental to epithelial barriers—associated with industrialization, urbanization, and modern lifestyles—contribute to the growing prevalence of allergic, autoimmune, and other chronic conditions. It is suggested that cellular content released from damaged barrier cells and the translocation of microbiota play a role in initiating inflammatory responses [144].

Historical evidence suggests that certain immune-mediated diseases have emerged in parallel with these microbial changes. For example, symmetrical polyarticular RA and similar types of juvenile idiopathic arthritis (JIA) are considered relatively recent in Europe [145]. The pathogens of this infection-related immune disease could be transmitted from Colonial America to Europe in the 17th century and then spread slowly around the world, implicating strains within the human microbiota as potential contributors. The pathogenic strains in microbiota colonize the host first, and then they invade host or/and host cells on occasion, releasing disease-eliciting substances from infected and injured cells. Another example is KD, which appeared to be a newly emerging disease in East Asia countries including Japan, South Korea, Taiwan, and China prior to the 1950s [146]. KD has been present in Europe, referred to infantile polyarteritis nodosa as a severe form [147]. More recently, KD has been increasingly reported worldwide, a trend that seems to correlate with environmental changes such as economic development and adoption of westernized diets high in red meats [5,148]. The prevalence or incidence of acute and chronic infection-related immune-mediated diseases, including KD, MIS-C, JIA, inflammatory bowel disease, and Bechet’s disease, vary considerably across ethnic groups and populations. Interestingly, despite the variation in prevalence, the clinical manifestations of these diseases remain remarkably consistent, suggesting that immune functions against the disease are identical in individuals across different populations. This observation suggests that environmental factors such as distribution of pathogenic strains in microbiota play a more critical role than genetic factors [149]. Dysbiosis is now widely recognized as a contributing factor across multiple fields of medicine, including autoimmune, chronic inflammatory, psychological, neurodegenerative disorders, and cancer. Nevertheless, many challenges remain, especially in identifying the relationship between dysbiosis and disease onset [150,151].

The advance in genomics has enabled the successful sequencing of the human genomes, followed by those of various animal, microbial, and plant species genomes [152]. Traditional microbial culture techniques have limitations to detect certain pathogens, particularly in infection-associated conditions. To overcome these challenges, various genetic sequencing methods have been developed for detecting pathogens [153,154]. Genetic evidence has revealed that unculturable microbes are present in sites like the blood, placenta, synovial fluid, and even brain, which were once thought to be sterile [155,156]. Additionally, strains such as *Lactobacillus* and *Streptococcus* identified in human breast milk originate from the maternal gut, implying a route of endogenous microbial transmission [157]. The observations suggest that certain microbiota strains exist within the host and/or its cells, maintaining a dynamic and often symbiotic relationship. In addition, it has been observed that earlier highly pathogenic strains in microbiota such as influenza, scarlet fever, and acquired immunodeficiency syndrome tend to evolve toward less pathogenic strains over time. Similarly, infection-related immune-mediated diseases such as acute rheumatic fever, KD, Henoch-Schoenlein purpura (or IgA vasculitis), and childhood asthma has gradually become less severe and less frequent over time in South Korea [158,159,160].

External pathogens from environmental sources, such as *Legionella pneumoniae* and agents associated with sick building syndrome, and animal species-specific viruses such as avian influenza and SARS-CoV-1 viruses can affect human [161,162]. However, the diseases caused by these pathogens occur sporadically as isolated outbreaks and do not lead to cyclic epidemics and pandemics, due to the absence of asymptomatic carriers or reservoirs in the human species as non-human species-specific pathogens. In contrast, pathogens in microbiota, including human influenza virus, human corona viruses, and *Mycoplasma pneumoniae*, are maintained within the hosts in human species and are responsible for seasonal or periodic outbreaks with occasional pandemics. Also, some bacterial strains, including *S. pneumoniae*, *H. influenzae*, and *P. aeruginosa*, and fungi are harbored asymptomatically by healthy individuals, indicating the presence of persistent reservoirs in the human species [163].

Taken together, it is expected that many human diseases, including autoimmune, chronic inflammatory diseases and cancer, are associated with infection caused by microbiota strains, including unidentified and unculturable ones. However, most studies on this issue have focused on pathogens and PAMPs [164,165], not on the etiological substances within the self-cells. Although most allergens are considered external in origin, the etiological substances that trigger allergic reactions are associated with internal pathogen infections [166,167]. According to the PHS hypothesis, etiological substances responsible for clinical manifestations of infectious diseases, infection-related immune-mediated diseases, and possibly some allergic disorders originate from host cells that have been infected and injured by microbiota strains as discussed earlier [9,10].

## 10. Pathophysiology of Allergic and Autoimmune Diseases Under the PHS Hypothesis

Although allergic reactions are traditionally categorized into four major types of hypersensitivity: immediate, cytotoxic, immune complex, and cell-mediated types, some allergic reactions do not clearly align with these categories [168]. Similarly, pathogenesis of allergic diseases such as asthma and chronic rhinosinusitis is divided into two mechanisms; atopic or IgE-mediated and non-atopic or non-IgE-mediated, despite patients from both groups present similar symptoms. Despite this classification, all allergic diseases, like other immune-mediated conditions, involve specific etiological substances and corresponding immune components. It was suggested that asthma and allergic rhinitis represent the ‘united airway disease’, since some patients show similar immune responses in both diseases [169]. Also, the concept of ‘allergic march’ has been proposed, suggesting that children tend to develop a series of allergic conditions sequentially such as atopic dermatitis, allergic rhinitis, asthma, and food allergy, though not necessarily in a fixed order [170]. However, the target cells affected by each etiological substance differ between the upper and lower respiratory tracts, and the immune status of allergic persons against the same allergens also vary. As a result, the substance responsible for symptoms and the corresponding immune responses differ among patients, even within the same allergic disease category. Furthermore, the immune status of children evolves with age, and many cases of infantile asthma, childhood AD, and food allergies show significant improvement over time [171].

External complex allergens contain toxic chemicals and biochemicals, and these small substances affect target cells via receptors expressed on the surface or in the cells. When the burden of such substances exceeds the capacity of the immune system, severe reactions, even fatal outcomes, occur in all individuals, regardless of allergic status. In common allergen exposure, the immune system of allergic people may have stronger memory and cross-reactivity abilities due to lack of immune effectors against external exposure, compared with non-allergic people as previously discussed. In addition, it is important to recognize that the current immune system should be understood to include not only traditionally known immune components, but also additional immune factors that control the non-protein/non-peptide substances. This encompasses toxic substances that remain unidentified, for which the associated immune mechanisms have not yet been characterized.

In general, allergic reactions are temporary, recurrent, and reversible, and most etiological substances are external origins and directly invade the host. Therefore, eosinophils, mast cells/basophils, and IgE are primary effectors against these external substances. Allergic reactions can be triggered not only by complex allergens but also by isolated chemicals, biochemicals, and emotional upset. Thus, allergic diseases comprise a range of heterogenous phenotypes including food allergies, drug allergies, occupational allergies, and exercise induced bronchial constriction (or asthma). In contrast, autoimmune diseases are typically characterized by sustained cell destruction, often driven by intracellular substances of internal origin, and T cells and B cells act as the principal immune effectors. Therefore, when cellular damage occurs as a result of any allergic responses, adaptive immune responses become activated. If these toxic substances are not properly controlled, chronic clinical symptoms resembling autoimmune disorders occur in patients with allergic diseases (Figure 1).

Symptoms and signs in allergic disease are triggered by etiological substances that are responsible for clinical manifestations and target cell injury. Toxic substances originating from external or less common internal allergens encompass a range of molecular materials, each characterized by unique sizes and biochemical properties. The toxic or bioactive substances bind to affinitive receptors of the target cells in various organ tissues, for instance, respiratory tract cells in asthma and skin cells in atopic dermatitis. Signaling from affected target cells recruits and activates immune effectors, causing localized inflammation. Depending on the size and biochemical property of the substances, known immune components in allergic responses such as eosinophils, mast cells/basophils and IgEs, and other immune components, including T cells and B cells, are activated for control of the toxic substances (see Table 1). Free or isolated allergens can directly activate corresponding immune cells through binding to receptors of immune cells. Eosinophils and mast cells control the toxic substances through their recognizing receptors expressed on the cells and corresponding immune effectors within the cells. IgEs control pathogenic proteins that invade directly from outside and those bound on target cells such as mast cells. Mast cells located in the CNS contribute to allergic responses by interacting with peripheral mast cells via the mast cell-associated network. The pathogenic proteins/peptides and other substances released from injured target cells following allergic reactions activate both adaptive and innate immune components for control of the substances. These reactions occur in a localized lesion (as the focus) first, and then inflammation-inducing mediators produced at the focus can affect target cells in near and remote organ tissues through systemic routes in allergic contexts such as asthma, anaphylaxis, and food allergies. In allergic patients, interaction between the toxic substances and corresponding immune components primarily occurs through cross-reactivity due to insufficient effective immune effectors. This interaction leads to cytokine imbalance and increased production of inflammatory mediators from immune cells, which contribute to clinical manifestations and target cell injury observed in allergic diseases.

### 10.1. Asthma

Asthma is a well-known chronic disease with historical records dating back to the time of Hippocrates. Asthma began to draw increased medical attention during the Industrial Revolution, likely due to a rise in cases associated with deteriorating air quality and the onset of modern science. Today, asthma is recognized as a heterogenous condition containing multiple diseases, with distinct pathogenetic mechanisms, therapeutic responses, and clinical outcomes [172,173]. Although asthma presents various phenotypes, it often develops during childhood, when the immune system, including ‘psychological system’ of the CNS, are immature. Since acute asthma attack can be triggered by psychological stimuli, it was once proposed that the ‘intrinsic asthma’ might be linked to psychological factors [174]. In fact, patients with asthma tend to have a higher incidence of comorbid psychological disorders such as anxiety, depression, and chronic stress [175,176].

The basic pathophysiology of acute asthma is the reversible constriction of the bronchioles on both sides of the lungs, resulting in symptoms such as wheezing, dyspnea, and chest tightness. This is supported by radiographic findings showing the symmetrical hyperinflation of both lungs. The substances that constrict the bronchial muscles reach the target cells in the lungs through systemic circulation, and there is a source producing these substances. In infants, acute bronchiolitis is caused by a variety of respiratory viruses, especially respiratory syncytial viruses and affects the bronchioles of both lungs, like asthma. Symptoms such as dyspnea and wheezing possibly occur due to mucus obstruction caused by narrow diameter of infant bronchioles [177]. These bilateral and symmetrical lesions are not due to direct viruses themselves, but rather to inflammation-inducing substances from initial sites of virus-infected cells (the focuses). In allergic asthma, for example, inflammatory mediators released from activated eosinophils or mast cells at the initial site of allergen exposure circulate systemically and reach the target cells of lungs, potentially triggering bronchial inflammation, constriction, or mucus obstruction. The mast cell-related signaling pathways in the CNS (the mast cell-associated network) also contribute to this process in both extrinsic and intrinsic asthma. In addition, when the allergens contain toxic substances that stimulate IgE, the resulting IgE-mediated responses exacerbate inflammation. If injury of target cells occurs due to allergens or allergic events, pathogenic proteins and peptides derived from injured cells activate neutrophils, T cells, especially Th2 cells, and B cells. It was proposed that airway epithelial cells injured during asthma exacerbations release alarmins and other mediators that promote a Th2 dominant inflammatory response [135]. Although most patients recover from acute allergic episodes, repeated immune activation and tissue injury lead to structural remodeling of the airways or the development of nasal polyps as a result of ongoing cycles of inflammation and repair [178,179].

Beyond the classification of asthma into phenotypes and endotypes, severe asthma remains a clinical challenge, as is the case with many chronic diseases. Most severe exacerbations of asthma are associated with infection or activation of underlying diseases. In these cases, the primary focus, that is, the localized lesions of injured cells by insults from infections or activation of underlying diseases, produce excess inflammation-inducing substances, and these substances attached to target lung cells through systemic route, manifesting severe pneumonia or bronchiolitis with wheezing and respiratory distress. Furthermore, the injured target lung cells such as epithelial or endothelial cells can also produce inflammatory substances that affect adjacent cells or even distant organ cells. Broken respiratory barriers allow not only easier access for environmental allergens but also for invasion of colonized microbial strains in the lower respiratory tract into the host, resulting in further immune activation such as septic conditions. If the inflammatory substances are not rapidly controlled by the immune system, the condition persists and worsens. In such scenarios, some asthma patients experience the delayed recovery process when accompanied by pneumonia or exacerbation of underlying diseases. This can lead to chronic lung inflammation like the pathophysiology of autoimmune diseases. Also, some patients with allergic asthma are affected with comorbidities such as chronic obstructive pulmonary disease (COPD) and other chronic pulmonary disorders, which lead to poorer clinical outcomes.

### 10.2. Atopic Dermatitis

The first clinical symptom of AD begins with the pruritus of the skin. Etiological substances such as natural toxins, irritants, and toxins from ingested food bind to and stimulate sensory nerve endings and possibly mast cells. The stimulation triggers itching through efferent signaling from the CNS. It has been suggested that the skin cells in patients with AD have genetic variations responsible for impaired skin barrier function, manifesting as dry skin and increased susceptibility to pruritus upon contact with allergens or internal triggers such as sweat [180]. Physical injury of skin cells by scratching or allergen (toxin)-induced cytopathy elicits releasing intracellular substances, which induce activation of adaptive and innate immune cells. Secondary bacterial invasion caused by broken skin barriers further amplifies the immune responses. In addition, the unregulated penetration of external allergens, including etiological substances, leads to increased activation and accumulation of IgEs, eosinophils, T cells, neutrophils, and other immune components in affected skin legions. In AD, the immune system cannot effectively regulate initial etiological substances and subsequently released substances from damaged skin cells, leading to chronic inflammation and a vicious symptom cycle. Notably, some patients, especially children, exhibit symmetrical lesion distribution, and emotional stress also exacerbates symptoms of atopic lesions, suggesting that certain etiological substances can act through systemic routes, including activation of the mast cell-associated network and psychological regulatory circuits of CNS, similar to mechanisms proposed in ‘intrinsic asthma’.

### 10.3. Other Allergic, Chronic Inflammatory and Autoimmune Diseases

Although a comprehensive discussion of all allergic, autoimmune, and chronic inflammatory conditions, including systemic autoinflammatory diseases and amyloidosis, is beyond the scope of this article, their pathophysiology may be conceptually unified under the framework of the PHS hypothesis.

Autoimmunity such as the appearance of autoantibodies often occurs during or after infectious diseases but occurs in healthy people. This suggests that autoantibodies occur after the cell damage caused by various pathological and non-pathological conditions and serve additional biological functions [181,182]. According to the PHS hypothesis, the pathophysiology of autoimmune and chronic inflammatory diseases is similar, with primary distinction being a higher frequency of autoantibodies and potentially self-reactive T cell clones in autoimmune diseases. Patients with allergic diseases or chronic lung diseases, such as asthma and COPD can also exhibit autoantibodies [183,184]. Conversely, patients with systemic autoimmune diseases such as RA, SLE, and systemic sclerosis, are often affected by pulmonary involvement including pneumonia, asthma, and chronic lung diseases [185]. Besides asthma as one of lung specific disorder, there are subacute or chronic non-infectious lung diseases, including hypersensitivity pneumonitis, chronic inflammatory lung diseases, sarcoidosis, and idiopathic pulmonary fibrosis. It has been suggested that a considerable proportion of these cases are attributable to infectious insults, particularly those caused by intracellular pathogens such as mycobacteria, fungi, or unidentified organisms [166,186,187,188].

Researchers have investigated various intracellular or systemic biological processes, including inflammasome and pyroptosis, apoptosis, autophagy, proteosome, thromboembolic pathway, glycolysis metabolism, and epigenetic modifications such as microRNAs regulation and DNA methylation. Numerous biomarkers derived from these studies have been shown to be activated or changed during disease progression. Certain biological changes serve as diagnostic or prognostic biomarkers and be used to develop targeted therapies for each disorder. On the other hand, it remains unclear whether such biological alterations directly contribute to disease development and progression, or they represent the secondary or adaptive response for recovery of the disease. It is reasonable to assume that specific etiological substances act as initial triggers of diverse biological alterations under each disorder. The clinical phenotypes, including those involving specific organs, are determined by the nature and amount of the etiological substances, their target cells within the affected organs, and the host’s ability to elicit effective (or specific) immune components to both the initial triggers and secondary substances derived from injured cells. The pathophysiology of other acute and chronic diseases affecting specific organs could be explained through similar mechanisms.

## 11. Treatment

Avoiding allergens is a fundamental strategy in managing allergic diseases. However, complete avoidance is often difficult or unfeasible for long-term disease control.

A wide range of therapeutic approaches have been developed for rapid symptom-relief and prevention of exacerbation of allergic diseases, including medications, allergen specific immunotherapy, and psychological approaches [172]. Since allergic, autoimmune, and chronic inflammatory diseases are involved with hyper- or abnormal immune responses, immune modulators such as corticosteroids (CSs) are commonly used for acute symptom management and long-term control. Additional immune modulators include intravenous immunoglobulin (IVIG), immune-suppressive chemicals such as methotrexate and cyclophosphamide, and biologics targeting specific cytokines or their receptors. The biologics act on molecules including interleukin-1 (IL-1), IL-4, IL-6, IL-13, and tumor necrosis factor-α (TNF-α) [189,190,191]. Each treatment acts through distinct mechanisms to reduce inflammation by interrupting various immune signaling pathways. As previously discussed, since inflammation is dynamic interaction between toxic substances and corresponding immune components, acute inflammation is often driven by nonspecific immune components together with cytokine imbalances before appearance of specific ones, whereas chronic inflammation is partially maintained by relevant nonspecific immune components because of lack of specific ones. Among immune-modulators, CSs (as analogs of cortisol) and IVIG (as serum IgG) can be regarded as host-derived immune regulators in vivo. It is possible that during acute immune episodes, a host immune system cannot produce enough of these regulators within the short time following exposure to excessive load of toxic substances. CSs may induce early immune stability through control of hyperimmune reactions occurred by nonspecific, cross-reactive immune components and affect nonspecific adaptive and innate immune cells [11]. Thus, CSs are effective in diseases where pathophysiology involves activation of nonspecific, immature, and/or cross-reactive B cells, T cells, as well as innate immune cells such as eosinophils and possibly mast cells, commonly observed in allergic and autoimmune diseases. Systemic CSs are known to rapidly decrease lymphocytes and eosinophils, often resulting in an immediate improvement of clinical symptoms especially in acute episodes. However, in the immune-mediated diseases which are associated with the immune-protein systems acting against idiopathic nephrotic syndrome, amyloidosis, and Alzheimer’s disease, the efficacy of CSs may be delayed or even absent [12].

Although CS is centered in the treatment of most immune-mediated diseases, chronic use of CS is associated with significant systemic adverse effects. These include Cushing face and growth retardation in children, osteoporosis, obesity and diabetes, peptic ulcer, hypertension, and neuropsychiatric complications [192]. Therefore, the long-term administration of CSs necessitates appropriate monitoring strategies. Allergic and autoimmune diseases are heterogeneous disease spectra, encompassing diverse phenotypes from asymptomatic (or in remission) to potentially fatal cases. Individuals with each disease have different clinical courses with remission time. Thus, the guidance for treatment of each immune-mediated disease has recommended stepwise approaches according to phenotypes and severity to minimize the long-term use of systemic CSs. In allergic diseases, the use of localized CS formulations such as inhaled CSs for asthma and topical CSs for AD, is developed to limit systemic exposure. However, long-term use of high-dose inhaled CSs can induce systemic side effects [193], indicating that the drugs can be absorbed via respiratory mucosa and then act systemically. To understand the pathophysiology of each allergic disease is necessary for proper control of acute exacerbation and long-term morbidity. For instance, in certain cases of asthma triggered by viral infections such as influenza or COVID-19, some asthmatic patients develop rapid progressive pneumonia and/or ARDS. This type of asthma does not response to inhaled CSs and conventional dose of systemic CS in asthma. Such cases are related to cytokine storms caused by excessive activation of non-specific immune cells releasing proinflammatory cytokines and proteolytic enzymes that exacerbate lung tissue injury. Therefore, early and aggressive treatment with high-dosed systemic CSs and/or IVIG, is crucial to reduce morbidity and prevent permanent lung injury during the early stages of the disease, especially in immune-competent patients [8,9,11]. This therapeutic strategy provides a rationale for initiating prompt immune modulator treatment in acute or subacute whole-organ destructive conditions, such as ARDS, myocarditis, rapidly progressive glomerulonephritis, fulminant hepatitis, necrotizing pancreatitis, adrenal necrosis, extensive epidermolysis, acute encephalopathies, and rapidly progressive RA and SLE. Certain asthma patients can be in comorbid chronic lung diseases, such as COPD, pulmonary fibrosis, and asthma-COPD overlap syndrome [194]. In these patients, immune proteins, including IL-1, IL-6, and TNF-α, play roles in continuous inflammation due to underlying immune dysregulation. Blocking one of immune proteins can reduce inflammation and maintain remission, as observed in autoimmune diseases. Therefore, the immunomodulators used for autoimmune diseases can be effective in reaching a remission state for patients with severe and intractable allergic diseases, who may have unidentified chronic immune-mediated conditions.

Both allergic diseases and autoimmune diseases represent constitutional disorders influenced by individual immune status determined by genetic factors. Immunomodulators such as CSs, biologics, and anti-inflammatory medications can alleviate symptoms and induce remission, but they are unlikely to achieve a complete cure since these therapies cannot resolve etiological substances of the disease. Thus, long-term prognosis of the diseases depends on the host’s capacity to control the etiological substances occurred during disease process.

Immunotherapy for allergic diseases has been used in clinical fields [195]. The specific etiological substances in allergens and corresponding immune components vary among individuals and are not clearly defined; thus, the efficacy of immunotherapy may be difficult to predict on an individual basis. Notably, a substantial proportion of patients with allergic diseases such as asthma, eczema, and food allergy in childhood and autoimmune diseases such as some phenotypes of JIA and SLE demonstrate self-limited or milder clinical courses over time. It suggests that with maturation of the immune system, some individuals acquire an immune system in which etiological substances are controlled properly over time and possibly during immunotherapy or other treatments. Conversely, the decline in immune system with aging referred to as immunosenescence can contribute to the emergence of asthma and other immune-mediated diseases such as neurodegenerative disorders and cancer [12,13,196].

Psychological insults have been associated with exacerbation of allergic diseases. Intervention such as cognitive behavioral therapy was reported to be helpful for long-term management of allergic diseases [197]. Psychological assistance may be effective especially in children who are more susceptible to psychological influences. Treatment strategies should be considered to reflect not only the pathophysiology of the disease but also the individual’s immune status with respect to age and comorbidities, as well as socioenvironmental factors and disease severity. Recently, it was proposed that management strategies for RA might take a more holistic approach, beyond merely therapeutic efficacy, with consideration of each patient’s multimorbidity profile and preferences, as well as the safety profile of available treatments [198]. These strategies may be particularly useful for individuals with severe and ‘difficult to treat’ allergic diseases.

## 12. Conclusions

Allergic diseases represent heterogeneous diseases with diverse etiologies and pathophysiological mechanisms, many of which remain incompletely understood. The immunopathogenesis of diseases, including allergic diseases, is explained through an established immunological dogma emphasizing the central role of T cells in immune regulation. However, such frameworks have limitations to explain the complex nature of allergic diseases. Immune cells and immune proteins from adaptive and innate immune systems are involved in host defense against not only infections but also noninfectious conditions including allergy, autoimmunity, trauma (or wound healing), transplantation rejection, intoxication, and cancer. The immune system controls smaller substances that originate from infectious agents and those derived from self-cells injured by various insults. Under the PHS hypothesis, all diseases involve etiological substances, and each immune component recognizes and controls these toxic substances, based on their size and biochemical properties for protecting host cells. Therefore, allergic diseases also have etiological substances that trigger immune responses, leading to clinical manifestations and injury to target cells. The substances are typically external in origin and are often small molecules such as chemicals, biochemicals, monoamines, neuropeptides, and proteins/peptides, including those bound to these materials acting as haptens. They exist as complex allergens or as free materials, each possessing distinct biologically active properties. In allergic responses, eosinophils and mast cells control the toxic substances through recognizing receptors expressed on the cells and corresponding immune effectors within the cells. IgE controls pathogenic proteins that invade directly from outside and those bound on target cells such as mast cells. Mast cells within the CNS may be associated with allergic responses through their communication with peripheral mast cells (i.e., the mast cell-associated network). The toxic proteins, peptides, and other substances released from injured cells following allergic reactions activate both adaptive and innate immune components to control the substances. The clinical phenotype of allergic diseases as well as other diseases is determined by the origin and nature of etiological substances, affected target cells in specific organs, and the corresponding immune effectors to those substances. Through the PHS hypothesis, we have provided the rationale of the early application of immune modulators (CSs and IVIG) in acute infectious diseases, including influenza, *Mycoplasma pneumoniae* pneumonia, and COVID-19, and infection-related immune-mediated diseases including KD, MIS-C, and other acute organ-specific diseases [8,9,10,11,199].

This article presents a novel perspective on the shared immunopathogenesis of allergic diseases under the PHS hypothesis. We hope that this conceptual framework will provide researchers with new perspectives into the complex mechanisms underlying allergic conditions and support the development of more effective therapeutic strategies.

## Figures and Tables

**Figure 1 ijms-26-08358-f001:**
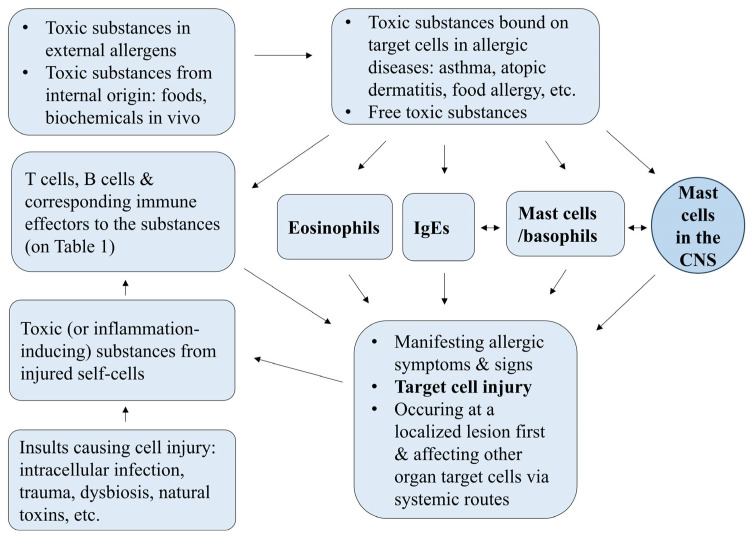
Schematic diagram of immunopathogenesis of allergic diseases. The arrows indicate the relationships across each content in the boxes. CNS stands for the central nervous system.

**Table 1 ijms-26-08358-t001:** Immune components in diseases, including allergies, under the PHS hypothesis.

Immune Components and Effectors (or Events)	Main Functions
Adaptive immune system	
T cells	Control of pathogenic peptides targeted to host cells, including cancer cells, through production of cytokines (possibly peptides) and immune effectors within the cells in the MHC-restricted (TCR-associated) and non-MHC-restricted events.
B cells	Control of pathogenic proteins targeted to host cells, including cancer cells, by production of antibodies in the MHC-restricted (BCR-associated) and non-MHC-restricted events.
Innate immune system	
Natural killer cells	Control of transformed cells such as virus-infected cells and tumor cells through their recognizing receptors and immune effectors within the cells.
Tissue macrophage-linaeged cells	Antigen presentation to adaptive immune cells in the MHC-restricted immune responses. Possible control of communications between immune cells and affected organ cells including cancers cells in TMEs [13].
Phagocytes (neutrophils and circulating monocyte/macrophages)	Control of large complex substances such as viruses, bacteria, parasites, and apoptotic & necrotic bodies associated with cell injury caused by infectin, trauma or other conditions.
Mast cells, basophils, eosionphils	These cells are activated by mainly external toxic substances that are recognized by receptors of the cells and control the substances through immune effetors within the cells and inflammation involved in these cells.
Unidentified innate immune components against small non-protein toxic materials	There are non-protein toxic or inflammation-inducing substances, including elements, monoamins, neuropeptides, LPS, RNAs, DNAs, chemicals and biochemicals. TLR-associated immune responses, natural antibodies, and immun proteins and/or peptides control these diverse substances. The immune proteins, including PrP gene products and amyloid proteins, control pathogenic monoamine metabolites or neuropeptides especially in CNS [12].
Production of alternative proteins in genetic diseases and cancer	The systemic and intracellular PHS control in part insults from a protein deficiency or malfunctional protein in organ tissues or within a cell [7,12,13].

MHC, major histocompatibility complex; TCR, T cell receptor; BCR, B cell receptor; TME, tumor microenvironment; LPS: lipopolysaccharide; TLR, Toll-like receptor; PrP, prion; CNS, central nervous system.

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
