# Peer review of "A Unified Pathogenesis of Allergic Diseases; The Protein–Homeostasis–System Hypothesis"

_ijms, 2025, doi:10.3390/ijms26178358_

Round 1
Reviewer 1 Report
Comments and Suggestions for Authors
1. The manuscript is excessively long and places so much emphasis on historical, general immunology background that the focus on the novel application of the PHS hypothesis for allergic diseases is boiling.
Specific suggested revisions: Keep the historical review and general immunology short. I think you should focus to allergic disease mechanisms as they relate to the PHS conceptual framework in the main text.
2. A clear definition linking to biochemical classes (protein, peptide, lipid and small molecule), size spectrum and potential ways of detection must be given right at the beginning of Introduction section. Use terminology consistently.
3. Many important concepts in this review (mast cell–CNS network, selective IgE deficiency and cancer risk, contribution of otherwise unidentified innate immune components) are proposed without direct evidence.Either cite additional peer-reviewed primary literature from independent groups to support these claims, or explicitly describe them as hypotheses which need investigative validation.
4. The text would benefit from visual representation. For example illustrate schematically the PHS framework and the individual components of it , the relationship among etiological agents, immune effectors and allergic clinical expressions, the hypothesized mast cells-CNS connections.
5. While the discussion of allergens, IgE, eosinophils, mast cells is complete it does lack linking to established concepts in allergy research.Please include more citations from reputable allergy and immunology literature, particularly in areas where consensus exists (e.g., food allergy diagnosis, mast cell biology, eosinophilic disorders).
Comments on the Quality of English Language
The manuscript uses long and complex sentences. Professional language editing would be beneficial to improve clarity
Author Response
|
Thank you very much for taking the time to review this manuscript. Please find the detailed responses below and the corresponding revisions/corrections highlighted/in track changes in the re-submitted files. Comments 1: The manuscript is excessively long and places so much emphasis on historical, general immunology background that the focus on the novel application of the PHS hypothesis for allergic diseases is boiling. Specific suggested revisions: Keep the historical review and general immunology short. I think you should focus to allergic disease mechanisms as they relate to the PHS conceptual framework in the main text. |
|
Response 1: We thank the reviewer for the comprehensive review of the manuscript. We agree with the reviewer’s comment that this paper is rather long. This paper aims to introduce the PHS hypothesis and immunopathogenesis of allergic diseases for basic and clinical researchers (most of them may be immunologists), thus background information and unresolved issues in current immunology are necessary. In addition, each section in this paper contains unresolved issues, which are associated with comprehensive understanding of pathophysiology of allergic diseases and the PHS hypothesis focusing on etiological substances, especially self-cell origin. It is true that in all human diseases, there are etiological (or inflammation-inducing) substances and limited immunological models for resolving the etiology of the diseases. - We omitted some paragraphs and sentences for reduction of overall length with keeping of main point of this study. We hope for your kind understanding and accept the main point of this study.
|
|
Comments 2: A clear definition linking to biochemical classes (protein, peptide, lipid and small molecule), size spectrum and potential ways of detection must be given right at the beginning of Introduction section. Use terminology consistently. |
|
Response 2: Thank you for your valuable comments. The etiological substances in the PHS hypothesis are diverse and variable in size and biochemical property from elements and monoamines to large proteins and other biochemicals. Thus, it is difficult to express and define the size and detection method of the diverse materials. Instead of this work, we added the sentences regarding the etiological substances in the Introduction, and for the size of proteins and peptides in subsection 5-1. Functional consistency… in the revised manuscript. P1-2, 2nd paragraph, line 39-45, and P8 2nd paragraph, line 331-334 (with new ref.51) Comments 3: Many important concepts in this review (mast cell–CNS network, selective IgE deficiency and cancer risk, contribution of otherwise unidentified innate immune components) are proposed without direct evidence. Either cite additional peer-reviewed primary literature from independent groups to support these claims, or explicitly describe them as hypotheses which need investigative validation. Response 3: Thank you for your valuable comments. The PHS hypothesis explains the relationship between the etiological substances and corresponding immune responses for pathophysiology of the diseases. The PHS hypothesis is based on ‘deductive reasoning’ and in the view of organicism for a unified immunopathogenesis of the diseases. Thus, there are few studies on this issue, and we tried to collect references broadly in biomedical fields as well as immunology, that are associated with PHS hypothesis. - We carefully revised sentences to avoid hypothetic auxiliary verbs such as “may or can “for description of published works. - Most studies in biomedical fields are based on reductive methods. However, reductive studies have also some problems. For example, if one suggests the mechanisms of cell injury in certain allergic diseases, he or she might explain them through the hypothesis based on results of the experiment with known immunological knowledge. The explanation may only be reliable in part, since the mode of action of studied immune component(s) or pathways is associated with other diverse components during the disease process and functions of majority of immune components involved are unknown. Comments 4: The text would benefit from visual representation. For example illustrate schematically the PHS framework and the individual components of it, the relationship among etiological agents, immune effectors and allergic clinical expressions, the hypothesized mast cells-CNS connections. Response 4: Thank you for your comments. The PHS hypothesis can be regarded as one of ‘theoretical biomedicine’ and the hypothesis is based on our clinical studies and endeavors of previous researchers in immunology and allergy as shown in this paper. The etiological substances and corresponding immune components in each disease are not defined at present time. Thus, detailed mechanisms in your suggestions remain for researchers in the future. If etiological substances and corresponding immune components are defined in the future, proper diagnosing, treatment, and prevention modalities will be developed in each disease. .- We corrected figure 1 and Table 1 with additional explanations for better understanding of the PHS hypothesis and allergic diseases. Comments 5: While the discussion of allergens, IgE, eosinophils, mast cells is complete it does lack linking to established concepts in allergy research. Please include more citations from reputable allergy and immunology literature, particularly in areas where consensus exists (e.g., food allergy diagnosis, mast cell biology, eosinophilic disorders). Response 5: Thank you for your comments. I understand your concerns on different concepts of allergic diseases in this paper. However, new theories are needed for development of immunology, allergology and biomedicine. - We rearranged and added some references from more reputable literature. (Ref. 17,18, 51,68,17, 109, 135 &144).
|
|
4. Response to Comments on the Quality of English Language |
|
Point 1: |
|
Response 1. We carefully revised sentences to avoid hypothetic auxiliary verbs such as “may or can “for published works and rewrote some long sentences to proper short sentences. |
Reviewer 2 Report
Comments and Suggestions for Authors
- Why do you broaden the PHS hypothesis to include cancer, genetic disorders, autoimmunity, and neurodegenerative diseases? This dilutes the emphasis on allergic mechanisms.
- You should consider exploring the epithelial barrier hypothesis, which posits that dysfunction of the epithelial barrier and microbial translocation can lead to the release of harmful protein-based substances that PHS would categorise and attempt to regulate. A comparative discussion could investigate potential synergies, such as how disruptions in the epithelial barrier might expose peptides or toxins that PHS mechanisms need to manage. Additionally, it could examine whether impaired barrier function overwhelms the capacity of PHS, potentially exacerbating disease.
- Figure 1 is too generic and does not include specific examples of etiological substances or immune pathways. Important concepts related to PHS are absent. Additionally, the visual hierarchy weakens the overall message. The arrows and labels do not effectively guide the reader through the logic of PHS. Moreover, there is no clinical correlation; the figure fails to connect mechanisms to real-world allergic phenotypes, such as asthma or eczema. A redesign is necessary.
- Lee, K.Y. is the most cited author in this paper's reference list, with 10 distinct publications referenced. While self-citations are valid for continuity, the paper could benefit from more independent validation of the PHS hypothesis. Only a few external studies directly support the idea of immune cells regulating protein homeostasis.
- There are no references provided in lines 438-450, 531-537, 563-573 and 613-628.
- In line 593, replace the incorrect word "proteostatsis" with the correct spelling.
Author Response
|
Thank you very much for taking the time to review this manuscript. Please find the detailed responses below and the corresponding revisions/corrections highlighted/in track changes in the re-submitted files. Comments 1: Why do you broaden the PHS hypothesis to include cancer, genetic disorders, autoimmunity, and neurodegenerative diseases? This dilutes the emphasis on allergic mechanisms. |
|
Response 1: We thank the reviewer for the comprehensive review of the manuscript. The PHS hypothesis could be regarded as one of the’ theoretical biomedicine’ and the hypothesis is based on our clinical studies and endeavors of previous researchers in immunology and allergy as shown in this paper. The PHS hypothesis simply explains the relationship between etiological substances and corresponding immune responses against them. Thus, it can explain the pathophysiology of all diseases, since all human diseases involve etiological substances, although to prove the hypothesis and its detailed mechanisms remain for researchers in the future.
|
|
Comments 2: You should consider exploring the epithelial barrier hypothesis, which posits that dysfunction of the epithelial barrier and microbial translocation can lead to the release of harmful protein-based substances that PHS would categorise and attempt to regulate. A comparative discussion could investigate potential synergies, such as how disruptions in the epithelial barrier might expose peptides or toxins that PHS mechanisms need to manage. Additionally, it could examine whether impaired barrier function overwhelms the capacity of PHS, potentially exacerbating disease. |
|
Response 2: Many thanks for your valuable suggestion. We introduced the epithelial barrier hypothesis, which is associated with the PHS hypothesis in section 9. Microbiota, Page 22, 2nd paragraph, line 978-983 (with reference 144). The etiological substances and corresponding immune components in each disease are not defined at present time. If etiological substances and corresponding immune components are defined in the future, more reliable pathophysiology and proper diagnosing, treatment, and prevention modalities will be developed in each disease. Comments 3: Figure 1 is too generic and does not include specific examples of etiological substances or immune pathways. Important concepts related to PHS are absent. Additionally, the visual hierarchy weakens the overall message. The arrows and labels do not effectively guide the reader through the logic of PHS. Moreover, there is no clinical correlation; the figure fails to connect mechanisms to real-world allergic phenotypes, such as asthma or eczema. A redesign is necessary. Response 3: Many thanks for your valuable suggestion. We corrected Figure 1 and Table 1 with additional explanations for better understanding of the PHS hypothesis and pathophysiology of allergic disease. Comments 4: Lee, K.Y. is the most cited author in this paper's reference list, with 10 distinct publications referenced. While self-citations are valid for continuity, the paper could benefit from more independent validation of the PHS hypothesis. Only a few external studies directly support the idea of immune cells regulating protein homeostasis. Response 4: I agree with your opinions. The PHS hypothesis is based on ‘deductive reasoning’ and in the view of organicism for a unified immunopathogenesis of the diseases. Thus, there are few studies on this issue, and we tried to collect references broadly in biomedical fields as well as immunology, that are associated with PHS hypothesis. - Most studies in biomedical fields are based on reductive methods. However, reductive studies have also some problems. For example, if one suggests the mechanisms of cell injury in certain allergic diseases, he or she might explain them through the hypothesis based on results of the experiment with known immunological knowledge. The explanation may only be reliable in part, since the mode of action of studied immune component(s) or pathways is associated with other diverse components during the disease process, and functions of majority of immune components involved are unknown. We hope for your kind understanding and accept the main point of this study Comments 5: There are no references provided in lines 438-450, 531-537, 563-573 and 613-628. Response 5: We added and rearranged references, and omitted sentences for the suggested parts, refs. 5&6, 51, 68, and 75, respectively, and added or changed some references 17,18,109,135,144 for improvement of the paper. Comments 6: In line 593, replace the incorrect word "proteostatsis" with the correct spelling Response 6: We correct the word.
|
Round 2
Reviewer 1 Report
Comments and Suggestions for Authors
Thank for submitting the revised manuscript. Authors addressed all the suggesting comments improving the clarity, quality and scientific rigor of the manuscript. I have no further concerns.
Reviewer 2 Report
Comments and Suggestions for Authors
Thank you for submitting your detailed point-by-point response to my comments on your manuscript.
The overall assessment is that your responses are thorough, accurate, and satisfactory. You have successfully addressed the majority of my concerns, with most suggestions being fully implemented. The revisions strengthen the manuscript's clarity, scholarly foundation, and overall argument.